# Closed-Form Test Functions for Biophysical Sequence Optimization Algorithms

**Samuel Stanton** [1]  **Robert Alberstein** [2]  **Nathan Frey** [1]  **Andrew Watkins** [2]  **Kyunghyun Cho** [1]

## Abstract

There is a growing body of work seeking to replicate the success of machine learning (ML) on domains like computer vision (CV) and natural language processing (NLP) to applications involving biophysical data. One of the key ingredients of prior successes in CV and NLP was the broad acceptance of difficult benchmarks that distilled key subproblems into approachable tasks that any junior researcher could investigate, but good benchmarks for biophysical domains are rare. This scarcity is partially due to a narrow focus on benchmarks which *simulate* biophysical data; we propose instead to carefully *abstract* biophysical problems into simpler ones with key geometric similarities. In particular we propose a new class of closed-form test functions for biophysical sequence optimization, which we call *Ehrlich functions*. We provide empirical results demonstrating these functions are interesting objects of study and can be non-trivial to solve with a standard genetic optimization baseline.

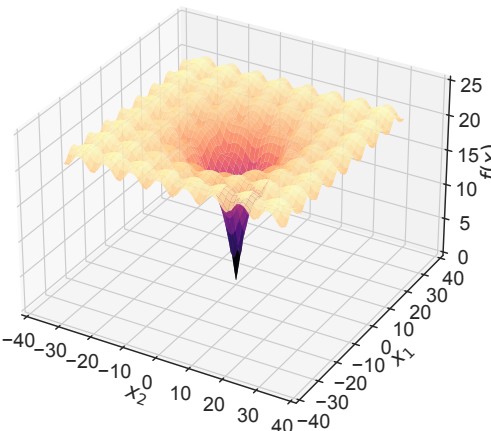

Fig. 1: The Ackley function is widely used to evaluate black-box optimization algorithms such as Bayesian optimization that have been successfully applied to many real-world problems. The relevance of the Ackley function is not its *semantic* correspondence with real-world objective functions, but its *geometric* similarities, such as a multiplicity of local minima and changing local curvature.

## 1. Introduction

Rigorous benchmarking is an essential element of good practice in science and engineering. Good benchmarks allow developers to evaluate new ideas rapidly in a low-stakes environment and thoroughly understand the strengths and weaknesses of their methods before applying them in costly, consequential settings. To see the benefit of a good benchmark, we need look no further than the Critical Assessment of Structure Prediction (CASP) competition (Bourne, 2003), which motivated AlphaFold (Jumper et al., 2021), or the many benchmarks in CV and NLP that shaped the development of modern deep learning (Russakovsky et al., 2015; Bojar et al., 2016; Hendrycks et al., 2020). While there has been a surge of investment into ML algorithms for ap-

plications like drug discovery, good benchmarks for those algorithms have proven elusive (Tripp et al., 2021; Stanton et al., 2022). Experimental feedback cycles in the life and physical sciences require expensive equipment, trained lab technicians, and can take months or even years. ML researchers require rapid feedback cycles, typically measured in minutes, necessitating proxy measures of success.

This need is particularly acute when evaluating black-box sequence optimization algorithms, which must produce a finite-length 1D sequence of discrete states (e.g. the primary amino-acid sequence of a protein or a segment of DNA) that optimizes a signal that is only accessible through measurements. Unlike typical ML benchmarks for supervised and unsupervised models, optimization algorithms cannot be evaluated with a static dataset unless the search space is small enough to be exhaustively enumerated and annotated with the test function. Many researchers turn to simulation or empirical function approximation to provide test functions for larger, more realistic search spaces, however there is always a compromise between the highest possible fidelity (which may still be quite imperfect) and acceptable

---

[1]Prescient Design, Genentech, New York City, USA [2]Prescient Design, Genentech, San Francisco, USA. Correspondence to: Samuel Stanton <stanton.samuel@gene.com>.

*Accepted at the 1st Machine Learning for Life and Material Sciences Workshop at ICML 2024.* Copyright 2024 by the author(s).

latency for rapid development.

Optimization algorithms are designed to solve *any* test function belonging to a certain class. For example, gradient descent provably converges (under suitable assumptions) to the global optimum of any differentiable convex function. The key observation is a test function need not correlate at all with downstream applications as long as there is shared structure (i.e. geometry). In fact, synthetic (i.e. closed-form) test functions (see Fig. 1) have been universally used to test continuous optimization algorithms for decades (Molga & Smutnicki, 2005).

In this work we ask, "What is the essential geometry of biophysical sequence optimization problems?" We draw inspiration from theoretical physics and biology, which rely heavily on analysis of tractable model systems to reveal interesting emergent behavior resembling empirical phenomena. For example, the Ising model is very simple, yet it predicts the existence of phase transitions and long-range order in atomic systems (McCoy & Maillard, 2012). Similar models have been used to predict protein folding pathways (Ooka & Arai, 2023) and study local mutational fitness landscapes (Neidhart et al., 2014), but have not been widely adopted by ML researchers. We propose *Ehrlich functions*[1] as an idealized model of real sequence optimization tasks like antibody affinity maturation, building on principles from structural biology and biomolecular engineering experience. Ehrlich functions have adjustable difficulty and are always provably solvable; easy instances can be solved quickly by a genetic algorithm and used for debugging, but the same algorithm fails to solve harder instances after consuming over 500M function evaluations. These results can be reproduced in minutes on a single GPU.

## 2. Benchmarking Principles and Approaches

We first briefly discuss the requirements of a good research benchmark, then general approaches to sequence optimization benchmarking in light of those requirements.

### 2.1. What Makes A Good Benchmark?

**Low costs/barriers to entry —** a good benchmark should be inexpensive and easy to use.

**Well-characterized solutions —** It should be easy to tell if a benchmark is "solved". Incremental progress towards better solutions should be reflected in the benchmark score.

**Non-trivial difficulty —** a good benchmark should be challenging enough to motivate and validate algorithmic improvements. It should not be possible to solve with naïve baselines on a tiny resource budget.

---

[1]Named after Paul Ehrlich, an early pioneer of immunology.

**Similarity to real applications —** while benchmarks inevitably require some simplification, a good benchmark should retain key characteristics of the desired application in a stylized, abstracted sense, otherwise the benchmark will not be relevant to the research community.

### 2.2. Existing Sequence Optimization Benchmarks

With these criteria in mind, we next categorize existing types of biophysical sequence optimization benchmarks. See Appendix A for further discussion of related work. While a robust benchmark should include a panel of test functions of varying types, we argue that closed-form functions are particularly useful to include and often overlooked.

**Database lookups —** database lookup test functions are constructed at substantial cost by exhaustively enumerating a search space and associating each element with a measurement of some objective, sometimes requiring large interdisciplinary teams of experimentalists and computationalists (Barrera et al., 2016; Wu et al., 2016; Ogden et al., 2019; Mason et al., 2021; Chinery et al., 2024). Unfortunately this approach necessarily restricts the search space, and the correctness of the database itself cannot be completely verified without repeating the entire experiment.

**Empirical function approximation —** empirical function approximation benchmarks are related to database lookups since they start from an (incomplete) database of inputs and corresponding measurements. This type of test function returns an estimate from a statistical model trained to approximate the function that produced the available data (e.g. hidden Markov model sequence likelihoods, protein structure models, or "deep fitness landscapes") (Sarkisyan et al., 2016; Rao et al., 2019; Angermueller et al., 2020; Wang et al., 2022; Verkuil et al., 2022; Xu et al., 2022; Notin et al., 2023; Hie et al., 2024). Unfortunately empirical approximation is only reliable locally around points in the underlying dataset, and it is difficult to characterize exactly over which region of the search space the estimates can be trusted. As a result, blindly optimizing empirical function approximators often reveals an abundance of spurious optima that are easy to find but not reflective of the solutions we want for the actual problem (Tripp et al., 2021; Stanton et al., 2022; Gruver et al., 2024).

**Physics-based simulations —** simulations are a very popular style of benchmark, but current options all violate different requirements of a good benchmark. Most simulations are slow to evaluate, many are difficult to install, some require expert knowledge to run correctly, and yet still in the end simulations can admit trivial solutions that score well but are not actually desirable. For example, docking models have been proposed as test functions (Cieplinski et al., 2023), but they do not have well-characterized solutions and are easy to fit with deep networks (Graff et al., 2021).

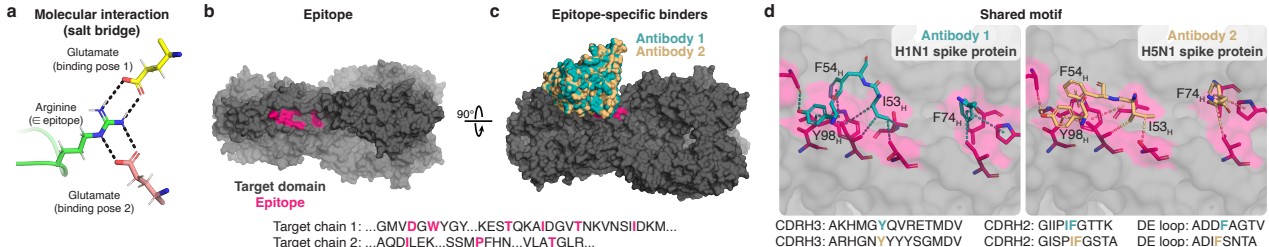

Fig. 2: **(a)** Arginine and glutamate are complementary amino acids because they have a strong *salt bridge* interaction. **(b - c)** Antibodies that bind to a specific region of a target protein (the *epitope*) have many therapeutic and diagnostic uses. **(d)** Antibodies with different sequences can bind to the same epitope on two homologous proteins because they are structurally similar, which manifests as shared motifs in sequence space. Structures shown have RCSB codes 3gbn and 4fqi.

The primary appeal of simulations is a resemblance to real applications, however the resemblance can be superficial. $\Delta\Delta G$ simulations (Schymkowitz et al., 2005; Chaudhury et al., 2010) do not have a low barrier to entry, and yet the correlation of $\Delta\Delta G$ with real objectives (e.g., experimental binding affinity) is generally modest or unproven (Kellogg et al., 2011; Barlow et al., 2018; Hummer et al., 2023). Despite their difficulties, simulation benchmarks can be an important source of validation for mature methods for which we can justify the effort. However the limitations of simulations makes them especially unsuited for rapid method development, leading us to explore other alternatives.

**Closed-form test functions** — closed-form functions have many appealing characteristics, including low cost, arbitrarily large search spaces, and amenability to analysis, however existing test functions for sequence optimization are so easy to solve that they are mostly used to catch major bugs. Simply put, designing a functioning protein is much, much harder than maximizing the count of beta sheet motifs (just one of many types of locally folded secondary structure elements in proteins) (Gligorijević et al., 2021; Gruver et al., 2024). The beta sheet test function highlights the main difficulty of defining closed-form benchmarks, namely not oversimplifying the problem to the point the benchmark becomes too detached from real problems.

## 3. Proposed Benchmark

Now we introduce Ehrlich functions, a closed-form family of test functions for sequence optimization benchmarking. In addition to defining the function class itself, we also explain which specific aspects of real biophysical sequence design problems are captured by this type of function, using antibody affinity maturation as a running example.

### 3.1. Uniform random draws uninformative

One of the first challenges encountered in black-box biophysical sequence optimization is a constraint on which se-

quences can be successfully measured. For example, chemical assays first require the reagents to be synthesized, and protein assays require the reagents be expressed by some expression system such as mammalian ovary cells. Popular algorithms like Bayesian optimization often assume the search space can be queried uniformly at random to learn the general shape of the function. If protein sequences are generated by stringing together uniformly random amino acids and sent to the lab, we learn nothing about the objective function (e.g. binding affinity) because the "proteins" do not fold into a well-defined structure and cannot be purified.

Unfortunately constraints like protein expression cannot currently be characterized as a closed-form constraint on the sequence, we only have examples of expressing proteins in databases like UniRef (Suzek et al., 2007). We simplify and abstract this feature of biophysical sequences with the notion of a feasible set of sequences $\mathcal{F}$ with non-zero probability under a discrete Markov process (DMP) with transition matrix $A \in \mathbb{R}_+^v \times \mathbb{R}_+^v$,

$$\mathcal{F} = \{\mathbf{x} \in \mathcal{X} \mid A[x_{\ell-1}, x_\ell] > 0 \; \forall \ell \geq 2\}, \qquad (1)$$

where $\mathcal{X} = \mathcal{V}^L$ is the set of all sequences of length $L \geq 2$ that can be encoded with states $\mathcal{V}$, with $|\mathcal{V}| = v$. Note that if sequences are drawn uniformly at random, then assuming at least one entry of $A$ is zero (i.e. at least one state transition is infeasible), we have

$$\mathbb{P}[\mathbf{x} \notin \mathcal{F}] \geq \sum_{\ell=1}^{L//2} \left(1 - \frac{1}{v^2}\right)^\ell \frac{1}{v^2},$$

$$= 1 - \left(1 - \frac{1}{v^2}\right)^{L//2},$$

where $//$ denotes integer division. If we choose $L$ large enough we will see uniform random draws fall outside $\mathcal{F}$ with high probability. See Appendix B.1 for further details on our procedure to generate random ergodic transition matrices with infeasible transitions.

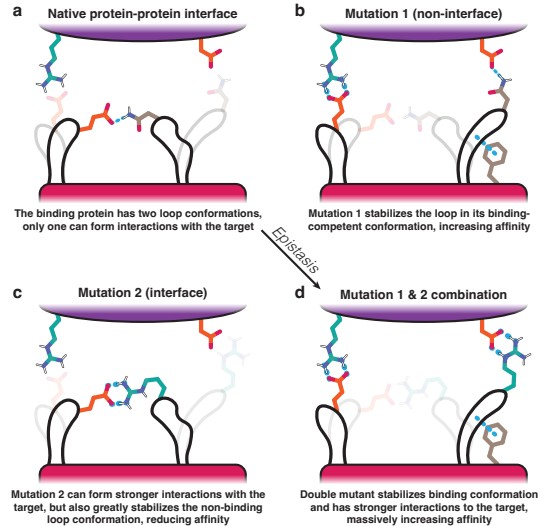

**a**    Native protein-protein interface

**b**    Mutation 1 (non-interface)

The binding protein has two loop conformations, only one can form interactions with the target

Mutation 1 stabilizes the loop in its binding-competent conformation, increasing affinity

*Epistasis*

**c**    Mutation 2 (interface)

**d**    Mutation 1 & 2 combination

Mutation 2 can form stronger interactions with the target, but also greatly stabilizes the non-binding loop conformation, reducing affinity

Double mutant stabilizes binding conformation and has stronger interactions to the target, massively increasing affinity

Fig. 3: Illustration of an epistatic second-order interaction.

### 3.2. Non-additive, position-dependent sensitivity to perturbation

By construction, any sequence optimization problem can be written as minimizing the minimum edit distance to some set of optimal solutions $\mathcal{X}^*$. In the antibody engineering context $\mathcal{X}^*$ is not a singleton but a set of solutions that all satisfy a notion of *complementarity* with the target antigen of interest (more specifically the target *epitope*). As a simple example, if the epitope has an arginine residue, then placing a glutamate residue on the antibody creates the possibility of a *salt bridge* (see Fig. 2). Furthermore, the formation of a salt bridge in this example requires that we place the glutamate at specific *positions* on the antibody sequence that are in contact with the epitope (i.e. on the *paratope*). One of the reasons there are many possible solutions to the antibody-antigen binding problem is the absolute position of an amino acid in sequence space can vary as long as the resulting structure is more or less the same (i.e. there are two or more *structural homologs*). The functional effect of changes to the antibody sequence are not only non-additive, but can exhibit state-dependent higher-order interactions, a phenomenon known as *epistasis* (Fig. 3).

We abstract these features of biophysical sequence optimization by specifying the objective as the satisfaction of a collection of $c$ *spaced motifs* $\{(\mathbf{m}^{(1)}, \mathbf{s}^{(1)}), \ldots, (\mathbf{m}^{(c)}, \mathbf{s}^{(c)})\}$, where $\mathbf{m}^{(i)} \in \mathcal{V}^k$ and $\mathbf{s}^{(i)} \in \mathbb{Z}_+^k$ for some $k \leq L//c$. Given a sequence $\mathbf{x}$, we can represent the degree to which $\mathbf{x}$ satisfies a particular $(\mathbf{m}^{(i)}, \mathbf{s}^{(i)})$ with $q \in [1, k]$ bits of precision

as follows:

$$h_q(\mathbf{x}, \mathbf{m}^{(i)}, \mathbf{s}^{(i)}) = \tag{2}$$

$$\max_{\ell < L} \left( \sum_{j=1}^{k} \mathbb{1}\{x_{\ell+s_j^{(i)}} = m_j^{(i)}\} \right) // \left( \frac{k}{q} \right) / q.$$

The quantization parameter $q$ allows us to control the *sparsity* of the objective signal (note that $q$ must evenly divide $k$). Taking $q = k$ corresponds to a dense signal which increments whenever one additional element of the motif is satisfied. Taking $q = 1$ corresponds to a sparse signal that only increments when the whole motif is satisfied. For example, if $k = 2$ and $q = 2$ then Eq. (2) can assume the values $0$, $0.5$, or $1$. If $k = 2$ and $q = 1$ then Eq. (2) can only assume the values $0$ or $1$.

We are now ready to define an *Ehrlich function* $f : \mathcal{V}^L \to (-\infty, 1]$, which quantifies with precision $q$ the degree to which $\mathbf{x}$ *simultaneously* satifies all $(\mathbf{m}^{(i)}, \mathbf{s}^{(i)})$ if $\mathbf{x}$ is feasible, and is negative infinity otherwise.

$$f(\mathbf{x}) = \begin{cases} \prod_{i=1}^{c} h_q(\mathbf{x}, \mathbf{m}^{(i)}, \mathbf{s}^{(i)}) & \text{if } \mathbf{x} \in \mathcal{F} \\ -\infty & \text{else} \end{cases}. \tag{3}$$

Note that we must take some care to ensure that 1) the spaced motifs are *jointly satisfiable* (i.e. are not mutually exclusive) and 2) at least one feasible solution under the DMP constraint in Eq. (1) attains the global optimal value of 1. See Appendix B.2 for details.

## 4. Experiments

In the last section we argued that Ehrlich functions capture important aspects of real-world sequence optimization problems, including antibody engineering. Now we show empirically that Ehrlich functions are indeed non-trivial to optimize, and demonstrate the type of experiments than we can conduct when the test function is inexpensive to evaluate. First we will introduce some key metrics we will use to assess optimizer performance, then we will show how different choices of Ehrlich function parameters can be manipulated to make test function more or less difficult for a fixed optimization algorithm to solve. Finally we will fix the Ehrlich function parameters and vary the optimizer hyperparameters to better understand the optimizer behavior.

### 4.1. Optimization Routine

We use a simple, robust genetic algorithm (GA) baseline (Back, 1996) to solve the black-box optimization problem $\max_{\mathbf{x} \in \mathcal{X}} f(\mathbf{x})$. See Appendix C.2 for pseudo-code and implementation details. We find that the following hyperparameters are fairly robust across different test function instances: $n = 10^6$, $\alpha = 10^{-4}$, $p_m = 1/L$, and $p_r = 1/L$.

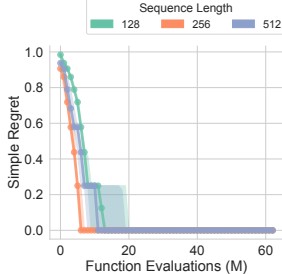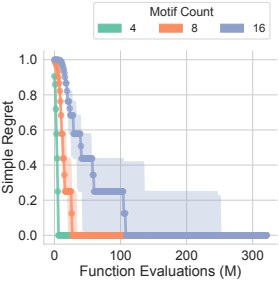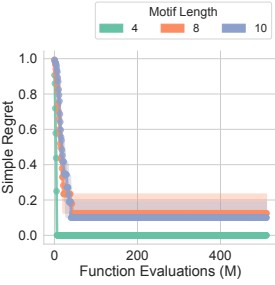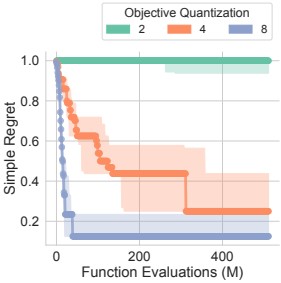

Fig. 4: Here we show how the difficulty of the test problem can be controlled by varying Ehrlich function parameters, keeping the optimizer fixed to a robust GA baseline. Starting from a fixed set of reference parameters we vary each parameter individually. For this optimizer, the problem difficulty depends most strongly on the quantization parameter $q$.

## 4.2. Optimizer Evaluation Metrics

In all plots we show the 10%, 50%, and 90% quantiles of the reported performance metrics, estimated from 32 trials.

**Regret —** let $\mathbf{x}^*$ be a global maximizer of $f$ that attains the optimal value $f^*$, and let $\widehat{\mathbf{x}_t^*}$ be an estimated maximizer at time $t \in [1, T]$ obtained by running some optimization algorithm for $T$ iterations. The simple and cumulative regret of the algorithm are respectively defined as follows:

$$r_t = f^* - f(\widehat{\mathbf{x}_t^*}), \quad R_t = \sum_{j=1}^{t} r_j.$$

**Feasibility —** genetic algorithms maintain a population of solutions (i.e. particles) $\mathcal{X}_{\text{pop}}$ which are randomly recombined, mutated, and evaluated with the test function. In addition to regret, another important metric is the feasible fraction of $\mathcal{X}_{\text{pop}}$ under the DMP constraint at iteration $t$,

$$\frac{1}{|\mathcal{X}_{\text{pop}}^{(t)}|} \sum_{\mathbf{x} \in \mathcal{X}_{\text{pop}}^{(t)}} \mathbb{1}\{\mathbf{x} \in \mathcal{F}\}.$$

## 4.3. Controlling Optimization Problem Difficulty

In Fig. 4 we demonstrate the effect of varying the Ehrlich test function parameters, keeping the optimizer hyperparameters fixed. We start from a base configuration $L = 256$, $c = 4$, $k = 4$, $q = k$ and vary one parameter at a time. Increasing $c$ and $k$ predictably increase the difficulty of the problem, as more function evaluations are required to find a global maximizer (i.e. a solution with 0 simple regret). Varying the sequence length only makes the problem more difficult up to a point, after which the problem becomes easier. We conjecture that medium length sequences are most difficult because there are many positions the optimizer can change, but the sequence is short enough that motif clashes are still difficult to avoid.

In the rightmost panel we fix $k = 8$ and vary $q$, which strongly affects the performance of the genetic optimizer.

Performance significantly degrades between $q = 8$ and $q = 4$, and totally collapses when $q = 2$. We know if $p_m = p_r = 1/L$ then in expectation the optimizer only searches 1-2 edits away from current solutions, which means the optimizer cannot search deep enough to find any solutions that would cause the objective value to increase, since for $k = 8$ and $q = 2$ we know those solutions could be up to 4 very specific edits away. In the next section we verify this intuition by optimizing the algorithm hyperparameters for this particular test instance and inspecting the results.

## 4.4. Understanding and Improving Optimization Algorithms

We fix the parameters of the Ehrlich test function to $L = 256$, $c = 4$, $k = 8$, and $q = 4$ and use the default Weights & Biases Bayesian optimization agent to search for better optimizer hyperparameters, minimizing cumulative regret. We evaluate 512 different hyperparameter configurations, consuming over 200B function evaluations in total. In Fig. 5 we compare the best hyperparameters found by this search (**A**: $\alpha = 2 \times 10^{-4}$, $p_m = 1.44 \times 10^{-2}$, and $p_r = 8.4 \times 10^{-3}$) to our starting values (**B**: $\alpha = 10^{-4}$, $p_m = 0.0039$, and $p_r = 0.0039$). Configuration **A** outperformed **B** on most random seeds, but some seeds show essentially no improvement. More aggressive mutations and recombination rates increase the depth of the search at each iteration, however they also drastically decrease the number of feasible solutions considered, wasting most of the function evaluations.

## 5. Discussion

In this work we have introduced a new closed-form family of test functions designed to capture key features of difficult biophysical sequence optimization problems. We have shown that these test functions can be used to define easy problems for debugging and hard problems that require millions of function evaluations to solve with a simple baseline. There are additional characteristics of real problems that could easily be incorporated, such as competing objectives,

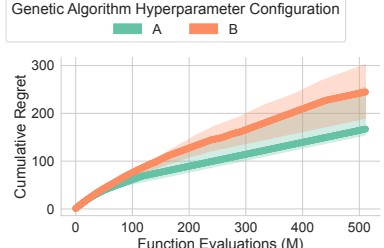
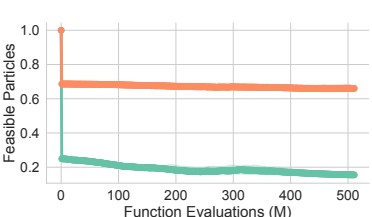

Fig. 5: Here we show the effect of tuning the GA algorithm hyperparameters to optimize a fixed Ehrlich function with $k = 8$ and $q = 4$. Configuration **A** is more aggressive than **B**, with higher values for $p_m$ and $p_r$. The optimal hyperparameter setting must trade off the depth of the search per iteration with the risk of violating the feasibility constraint.

observation noise, and environmental confounders, which we have chosen to omit because they do not fundamentally change the search problem.

In future work we intend to use this benchmark to thoroughly evaluate generative ML algorithms end to end for sequence optimization. Intuitively we can expect that this benchmark will help illuminate one of the key benefits of generative search, namely the ability to search deeper into sequence space in each iteration without sacrificing feasibility, a key advantage over uniform random search. We also hope our contribution will encourage other researchers to consider how they might distill their application into simple abstracted problems that can be easily studied by the broader research community, building more common ground of rigorous, easily reproducible empirical results.

## Acknowledgements

The authors thank Matthieu Kirchmeyer, Sidney Lisanza, and Clara Wong-Fannjiang for helpful feedback and discussion.

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

# A. Extended Related Work

There are a few notable efforts to improve the state of sequence optimization benchmarks for biophysical domains.

**Small molecules —** in the small molecule domain, GuacaMol (Brown et al., 2019) and the Therapeutics Data Commons (TDC) (Huang et al., 2021) include simulation-based test functions for small molecule generation/optimization benchmarking. As we discussed in the main text, simulation-based test functions have significant barriers to entry ranging from computational resource requirements to software engineering concerns such as dependency management. If these simulations were in fact well-characterized, high-fidelity proxies for real molecule design objectives then these objections could be resolved, however at the time of writing it is difficult to determine 1) when a simulated task is "solved" and 2) what constraints are required to prevent ML methods from "hacking" the simulation and 3) to what degree simulation scores correspond at all to actual experimental feedback. Indeed, one could argue that if real molecule design objectives were sufficiently well-understood to characterize via simulation then the most effective approach to ML-augmented molecule design would be to simply approximate and accelerate those simulations rather than directly model experimental feedback.

**Large molecules —** in the large molecule domain, ProteinGym (Notin et al., 2023) assembles a collection of protein datasets and model baselines but is primarily targeted at evaluating offline generalization with a fixed dataset. The models from this benchmark could be used as "deep fitness landscapes" (i.e. an empirical function approximation optimization benchmark), with the corresponding limitations we discussed in the main text. Our work is most closely related to the FLEXS benchmarking suite (Sinai et al., 2020).[2] To our knowledge, FLEXS is the most comprehensive attempt to date to assemble a robust suite of benchmarks for large molecule sequence optimization, with benchmarks for DNA, RNA, and protein sequences from an array of combinatorially complete database lookups, empirical function approximators, and physics simulators. Closed-form test functions are notably absent, hence our contribution can be seen as augmenting existing benchmark suites with test functions that are geometrically similar to real sequence optimization problems and also easy to install and cheap to evaluate.

## Models of Sequence Fitness in Theoretical Biology —

Geneticists have proposed theoretical models of biophysical sequence fitness and the geometry induced by random mutation and selection pressure, notably the mutational landscape model from Gillespie (2004), with more recent variants including the Rough Mt. Fuji model from Neidhart

---

[2]https://github.com/samsinai/FLEXS

et al. (2014). These models are interesting objects of study, however those models assume mutational effects are either independent or additive, which disagrees with the correlated non-additive structure we observe empirically. These models also do not account for "fitness cliffs" (i.e. expression constraints that are highly sensitive to local perturbation and determine whether function is possible to observe experimentally). We implemented the Rough Mt. Fuji model as an additional test function and verified that a genetic algorithm can easily optimize it. Ehrlich functions can be seen as a constrained, non-additive mutational fitness landscape, and may be interesting objects for further theoretical analysis.

# B. Constructing Ehrlich Functions

One major advantage of procedurally generating specific instances of Ehrlich functions is we can generate as many distinct instances of this test problem as we like. In fact it creates the possibility of "train" functions for algorithm development and hyperparameter tuning and "test" functions for evaluation simply by varying the random seed. However, defining a random instance that is nevertheless provably solvable takes some care in the problem setup, which we now explain.

## B.1. Constructing the Transition Matrix

Here we describe an algorithm to procedurally generate random ergodic transition matrices $A$ with infeasible transitions. A finite Markov chain is ergodic if it is *aperiodic* and *irreducible* (since every irreducible finite Markov chain is positive recurrent). Irreducibility means every state can be reached with non-zero probability from every other state by some sequence of transitions with non-zero probability. We will ensure aperiodicity and irreducibility by requiring the zero entries of $A$ to have a banded structure. For intuition, consider the transition matrix

$$\begin{bmatrix} 0.4 & 0.3 & 0 & 0.3 \\ 0.3 & 0.4 & 0.3 & 0 \\ 0 & 0.3 & 0.4 & 0.3 \\ 0.3 & 0 & 0.3 & 0.4 \end{bmatrix}$$

Recalling that $v$ is the number of states, we can see that every state $x$ communicates with every other state $x'$ by the sequence $x \rightarrow (x+1) \mod v \rightarrow \cdots \rightarrow (x'-1) \mod v \rightarrow x'$. We also see that the chain is aperiodic since every state $x$ has a non-zero chance of being repeated.

To make things a little more interesting we will shuffle (i.e. permute) the rows of a banded structured matrix (with bands that wrap around), but ensure that the diagonal entries are still non-zero. Note that permuting the bands does not break irreducibility because valid paths between states can be found by applying the same permutation action on valid paths from the unpermuted matrix. We will also choose the

non-zero values randomly, using the shuffled banded matrix only as a binary mask $B$ as follows:

$$
\begin{array}{c} \text{(banded matrix)} \\ \begin{bmatrix} 1 & 1 & 0 & 1 \\ 1 & 1 & 1 & 0 \\ 0 & 1 & 1 & 1 \\ 1 & 0 & 1 & 1 \end{bmatrix} \xrightarrow{\text{shuffle}} \begin{bmatrix} 1 & 0 & 1 & 1 \\ 1 & 1 & 1 & 0 \\ 1 & 1 & 0 & 1 \\ 0 & 1 & 1 & 1 \end{bmatrix}, \\[2em] \xrightarrow{\text{diag=1}} \begin{bmatrix} 1 & 0 & 1 & 1 \\ 1 & 1 & 1 & 0 \\ 1 & 1 & 1 & 1 \\ 0 & 1 & 1 & 1 \end{bmatrix} = B. \end{array}
$$

Now we draw the transition matrix starting with a random matrix with IID random normal entries, softmaxing with temperature $\tau > 0$ to make the rows sum to 1, applying the mask $B$, and renormalizing the rows by dividing by the sum of the columns after masking.

$$
\begin{array}{c} \text{(randn matrix)} \\ Z = \begin{bmatrix} +1.41 & +1.67 & -1.52 & +0.63 \\ -0.35 & +0.45 & +0.86 & -0.49 \\ +1.42 & -1.31 & -0.31 & +1.43 \\ -0.02 & +1.55 & -0.26 & +1.13 \end{bmatrix}, \\[2em] \xrightarrow{\text{softmax}} \begin{bmatrix} 0.36 & 0.46 & 0.02 & 0.16 \\ 0.13 & 0.30 & 0.45 & 0.12 \\ 0.44 & 0.03 & 0.08 & 0.45 \\ 0.10 & 0.49 & 0.08 & 0.33 \end{bmatrix}, \\[2em] \xrightarrow{\odot B} \begin{bmatrix} 0.36 & 0 & 0.02 & 0.16 \\ 0.13 & 0.30 & 0.45 & 0 \\ 0.44 & 0.03 & 0.08 & 0.45 \\ 0 & 0.49 & 0.08 & 0.33 \end{bmatrix}, \\[2em] \xrightarrow{\text{norm}} \begin{bmatrix} 0.66 & 0 & 0.04 & 0.30 \\ 0.15 & 0.34 & 0.51 & 0 \\ 0.44 & 0.03 & 0.08 & 0.45 \\ 0 & 0.55 & 0.09 & 0.36 \end{bmatrix} = A. \end{array}
$$

We can also verify that $A$ is ergodic numerically by checking the Perron-Frobenius condition,

$$
(A^m)_{ij} > 0, \ \forall i,j, \tag{4}
$$

where $m = (v-1)^2 + 1$, $A^1 = A$, and $A^b = A^{b-1}A$ for all $b > 1$. In our example, if $v = 4$ then $m = 10$ and we verify on a computer that

$$
A^{10} = \begin{bmatrix} 0.33 & 0.23 & 0.17 & 0.27 \\ 0.33 & 0.23 & 0.17 & 0.27 \\ 0.33 & 0.23 & 0.17 & 0.27 \\ 0.33 & 0.23 & 0.17 & 0.27 \end{bmatrix}
$$

## B.2. Constructing Jointly Satisfiable Spaced Motifs

Here we describe how to procedurally generate $c$ spaced motifs of length $k$ such that the existence of a optimal solution $\mathbf{x}^*$ with length $L$ with non-zero probability under a transition matrix $A$ generated by the procedure in Appendix B.1 can be verified by construction. If we simply sampled motifs completely at random from $\mathcal{V}^k$ we cannot be sure that a solution attaining a global optimal value of 1 is actually feasible under the DMP constraint.

First we require that $L \geq c \times k$. Next to define the motifs, we draw a single sequence of length $c \times k$ from the DMP defined by $A$ (the first element can be chosen arbitrarily). Then we chunk the sequence into $c$ segments of length $k$, which defines the motif elements $\mathbf{m}^{(i)}$. This ensures that any motif elements immediately next to each other are feasible, and ensures that one motif can transition to the next if they are placed side by side.

Next we draw random offset vectors $\mathbf{s}^{(i)}$. The intuition here is we want to ensure that an optimal solution can be constructed by placing the spaced motifs end-to-end. If we fix $c \times k$ positions to satisfy the motifs, there are $L - c \times k$ "slack" positions that we evenly distribute (in expectation) between the spaces between the elements of each motif. We set the first element of every spacing vector $s_1^{(i)}$ to 0, then set the remaining elements to the partial sums of a random draw from a discrete simplex as follows:

$$
\mathbf{w}^{(i)} \sim \mathcal{U}\left(\{\mathbf{w} \in \mathbb{R}^{k-1} \mid \sum w_i = 1\}\right). \tag{5}
$$

$$
s_{j+1}^{(i)} = s_j^{(i)} + 1 + \lfloor w_j^{(i)} \times (L - c \times k)//c \rfloor. \tag{6}
$$

Finally, recall that in Appendix B.1 we ensured that self-transitions $x \to x$ always have non-zero probability. This fact allows us to construct a feasible solution that attains the optimal value by filling in the spaces in the motifs with the previous motif elements.

As a concrete example, suppose $L = 8$, $c = 2$, and $k = 2$ (hence $s_{\max} = 3$) and we draw the following set of spaced motifs:

$$
\begin{bmatrix} 0 & 3 & 1 & 2 \end{bmatrix} \to \begin{bmatrix} 0 & 3 \\ 1 & 2 \end{bmatrix} = \begin{bmatrix} \mathbf{m}^{(1)} \\ \mathbf{m}^{(2)} \end{bmatrix}, \tag{7}
$$

$$
\begin{bmatrix} \mathbf{s}^{(1)} \\ \mathbf{s}^{(2)} \end{bmatrix} = \begin{bmatrix} 0 & 3 \\ 0 & 3 \end{bmatrix}. \tag{8}
$$

$$
\tag{9}
$$

An optimal solution can then be constructed as follows:

$$
\mathbf{x}^* = \begin{bmatrix} 0 & 0 & 0 & 3 & 1 & 1 & 1 & 2 \end{bmatrix}
$$

Note that this solution is only used to *verify* that the problem can be solved. In practice solutions found by optimizers

like a genetic algorithm will look different. Additionally if $L \gg c \times k$ then the spaced motifs can often be feasibly interleaved without clashes.

## B.3. Defining The Initial Solution

Optimizer performance is generally quite sensitive to the choice of initial solution. In our experiments we fixed the initial solution to a single sequence of length $L$ drawn from the DMP.

# C. Implementation Details

## C.1. Ehrlich Test Function Parameters

In addition to the parameters discussed in the main text, such as the sequence length $L$ and motif length $k$, there are a few additional parameters discussed in Appendix B that must be chosen. These parameters were fixed to the following values in all experiments

- Transition matrix bandwidth: $(v \times 2)//5$

- Transition matrix softmax temperature $\tau$: 0.5

## C.2. Genetic Algorithm Details

---

**Algorithm 1** Genetic algorithm pseudo-code

---

**Input:** initial solution $\widehat{\mathbf{x}^*}, \widehat{f^*}$, mutation probability $p_m$, recombination probability $p_r$, survival quantile $\alpha$, # particles $n$

$\mathcal{X}_{\text{pop}} \leftarrow \texttt{mutate}(\{\widehat{\mathbf{x}^*}\}, p_m, n)$
**for** $t = 1, \ldots, T$ **do**
$\quad \mathbf{v} \leftarrow f(\mathcal{X}_{\text{pop}})$
$\quad$ **if** $\max v_i > \widehat{f^*}$ **then**
$\quad\quad \widehat{\mathbf{x}^*} \leftarrow \operatorname{argmax} v_i$
$\quad\quad \widehat{f^*} \leftarrow \max v_i$
$\quad$ **end**
$\quad \tau \leftarrow \texttt{quantile}(\mathbf{v}, 1 - \alpha)$
$\quad \mathcal{X}_{\text{top}} \leftarrow \{\mathbf{x} \in \mathcal{X}_{\text{pop}} \mid f(\mathbf{x}) \geq \tau\}$
$\quad n' \leftarrow n - |\mathcal{X}_{\text{top}}|$
$\quad \mathcal{X}_{\text{pop}} \leftarrow \mathcal{X}_{\text{top}} \cup \texttt{recombine}(\mathcal{X}_{\text{top}}, p_r, n')$
$\quad \mathcal{X}_{\text{pop}} \leftarrow \texttt{mutate}(\mathcal{X}_{\text{pop}}, p_m, 1)$
**end**

**Returns:** Estimated maximizer $\widehat{\mathbf{x}^*}, \widehat{f^*}$

---

In Algorithms 1, 2, and 3, we provide pseudo-code for our genetic algorithm baseline, which we implement in pure PyTorch (Paszke et al., 2019), using the `torch.optim` API.

The GA baseline has only four hyperparameters, the total number of particles $n$, the survival quantile $\alpha \in (2/n, 1)$, the mutation probability $p_m$, and the recombination probability $p_r$. Generally speaking for best performance one should use the largest $n$ possible, and tune $\alpha$ (which determines the greediness of the optimizer), $p_m$, and $p_r$. However in practice it is not at all realistic to tune optimizer hyperparameters on the test problem, since there is usually little or no budget for tuning.

We find that for $n = 10^6$ (set by maxing out the memory of an NVIDIA A100 GPU in 32-bit precision), setting $\alpha = 10^{-4}$, $p_m = 1/L$, and $p_r = 1/L$ generally works well unless the objective function is "sparse", meaning that

---

**Algorithm 2** `mutate` function

---

**Input:** initial set $\mathcal{X}$, mutation probability $p_m$, number of mutants $n$.

$\mathcal{X}' = \emptyset$
**for** $\mathbf{x} \in \mathcal{X}$ **do**
$\quad$ **for** $i = 1, \ldots, n$ **do**
$\quad\quad \texttt{mask} = \texttt{rand\_like}(\mathbf{x}) < p_m$
$\quad\quad \texttt{sub} = \texttt{randint}(0, v - 1, \texttt{len}(\mathbf{x}))$
$\quad\quad \mathbf{x}' = \texttt{where}(\texttt{mask}, \texttt{sub}, \mathbf{x})$
$\quad\quad \mathcal{X}' = \mathcal{X}' \cup \{\mathbf{x}'\}$
$\quad$ **end**
**end**

**Returns:** $\mathcal{X}'$

---

**Algorithm 3** `recombine` function

---

**Input:** initial set $\mathcal{X}$, recombine probability $p_r$, number of recombinations $n$.

$\mathcal{X}' = \emptyset$
$\mathcal{P}^{(1)} = \texttt{draw\_w\_replacement}(\mathcal{X}, n)$
$\mathcal{P}^{(2)} = \texttt{draw\_w\_replacement}(\mathcal{X}, n)$
**for** $i = 1, \ldots, n$ **do**
$\quad \mathbf{x}^{(1)} = \mathcal{P}_i^{(1)}$
$\quad \mathbf{x}^{(2)} = \mathcal{P}_i^{(2)}$
$\quad \texttt{mask} = \texttt{rand\_like}(\mathbf{x}^{(1)}) < p_r$
$\quad \mathbf{x}' = \texttt{where}(\texttt{mask}, \mathbf{x}^{(1)}, \mathbf{x}^{(2)})$
$\quad \mathcal{X}' = \mathcal{X}' \cup \{\mathbf{x}'\}$
**end**

**Returns:** $\mathcal{X}'$

---

local perturbations of 1-2 edits usually cannot improve the objective value. More sparse objectives requires trading off the depth of the search in sequence space with the risk of violating the constraints, as we saw in our experiments in Section 4.4.

**Listing 1** Minimal optimizer benchmark code example

```python
import torch
from holo.test_functions import closed_form
from holo.optim import DiscreteEvolution

test_fn = closed_form.Ehrlich(negate=True)
params = [
    torch.nn.Parameter(
        test_fn.initial_solution().float(),
    )
]
optimizer = DiscreteEvolution(
    params,
    vocab=list(range(test_fn.num_states)),
    mutation_prob=1/test_fn.dim,
    recombine_prob=1/test_fn.dim,
    num_particles=1024,
    survival_quantile=0.01
)

for _ in range(4):
    loss = optimizer.step(
        lambda x: test_fn(x[0])
    )
```

## C.3. Code

Our code is available here: https://github.com/prescient-design/holo-bench. Listing 1 contains a minimal usage example.

We adopt the BoTorch API (Balandat et al., 2020) for the test functions, and the PyTorch `optim` API (Paszke et al., 2019) for the genetic algorithm. In addition to providing high levels of parallelization without the need for CPU multi-processing, we anticipate that our use of familiar APIs will make the baseline easier for the open-source community to use and draw clearer connections between continuous and discrete optimization algorithms.