# OpenReview forum: "Closed-Form Test Functions for Biophysical Sequence Optimization Algorithms"
_ICML.cc/2024/Workshop/ML4LMS — ML4LMS Poster_

### Official Review · Reviewer_2bq2 · 2024-05-29
**An insightful and intellectually honest paper centered around a smart idea.**

**Rating:** 9
**Confidence:** 4

**Review:**

This paper addresses a notable gap in biophysical sequence optimization research by introducing the Ehrlich functions, a new class of closed-form test functions. The authors propose a novel approach by abstracting biophysical problems into simpler ones with key geometric similarities, moving beyond the traditional focus on benchmarks that simulate biophysical data. The Ehrlich functions are demonstrated to be effective tools for studying sequence optimization, like antibody affinity maturation. The authors show that they can use the Ehrlich functions to define a range of problem sets, from simple ones suitable for debugging to complex ones requiring millions of function evaluations with a standard genetic optimization baseline.

Overall this paper is well written and well presented and displays a commendable versatility in both theoretical analysis and awareness of practical applications. The paper is candid yet thoughtful in its remarks on the current state of the research field. Overall I'm impressed by the novel ideas and the clear results presented in this paper. I do however have a few minor suggestion for improvement of the current manuscript:

- The plots are clean overall at but the size of the data-points is, in my opinion, too large to compared to the size of the figures and obscures the confidence bands, see for instance the 3rd panel of figure 4 and potentially the 3rd panel of figure 5.
- The description of the Ehrlich functions is rather complicated in its mathematical notation. While I actually enjoyed this, I wonder if it would be useful to provide pseudocode as an alternative, which might be more accessible to this particular community.
- The use of substructures in the Ehrlich functions reminds me of the (binary) fingerprints and the Tanimoto similarity used for small molecules, which makes me wonder if the Ehrlich functions (without the feasible requirement) could be used as a distance measure to compare existing peptides and proteins.

In conclusion, I found *Closed-Form Test Functions for Biophysical Sequence Optimization Algorithms* to be an insightful and intellectually honest paper centered around a smart idea. I can even go as far as saying that this has, up to now, been one of my favorite papers to read this year. Hence, I strongly recommend this paper for acceptance at the ICML 2024 Workshop ML4LMS.